# Lithium-Ion Cell Characterization, Using Hybrid Current Pulses, for Subsequent Battery Simulation in Mobility Applications

**Rares Catalin Nacu \* and Daniel Fodorean** 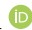

Department of Electrical Machines and Drives, Technical University of Cluj-Napoca,
400114 Cluj-Napoca, Romania
\* Correspondence: catalin.nacu@emd.utcluj.ro

**Abstract:** In this paper, a characterization method for a lithium iron phosphate (LFP) pouch cell is presented and evaluated, using a method that applies to hybrid current pulses called hybrid power pulse characterization (HPPC). The purpose of the study is to validate the developed mathematical model capable of offering good results for virtualization of the cell with extrapolation capability for the entire battery. This type of characterization was tested before but on cells with low capacity where relatively small currents were applied. Here, the model is intended to be used for the development of electrical mobility applications, such as electric vehicles (EV) and electric vehicle supply equipment (EVSE), where high capacity and currents are required through the cell. The comparison between the real and simulated cell was made with two sets of results obtained from HPPC and using the FTP-72 speed profile by emulating real current conditions, where both show that the method is reliable under the tested conditions and can be used for the considered application.

**Keywords:** hybrid pulse power characterization; electrical equivalent circuit; Li-ion cell; high-capacity cell; electrical vehicles; electrical vehicle supply equipment

## 1. Introduction

The number of charging points across EU, approximately 200,000, is still far from what is needed. Of those charging points, mainly due to grid limitations, only 1 in 7 has fast charging (>22 kW) capabilities [1,2]. One of the solutions is to use a battery within the station that behaves as an energy buffer between the grid and vehicle, reducing the grid load.

Thus, the construction of any type of EV and, in some cases, EVSE, requires the implication of a battery, but before that, it is vital for the entire system to be modeled and simulated. For this purpose, in order to be used in the system modeling, a battery cell needs to be characterized, simulated, and validated. There are a few characterization methods when it comes to a lithium iron phosphate (LFP) cell, considering different criteria as modeling perspectives (electrical model, electrochemical model, thermal model, mechanical model, or combinations as electro-chemical models), used equipment (potentiostat/galvanostat, source–load, climatic test chamber, shaker, etc.), levels (material level, cell level, pack level, system level), methods (sequence of pulses, electrochemical impedance spectroscopy), time scale of the models or characterization standards (IEC 62660-1, IEC 61982-4) [3]. For example, in [4], the authors evaluated the existing characterization methods for automotive applications with respect to the main international battery test standards, combining them into one procedure. Here, the HPPC current profile was used and the parameters were estimated for a RC parallel network followed by a RC series network; however, a comparison between the real and a simulated cell model is missing. Furthermore, in [5], the electrochemical impedance spectroscopy (EIS) technique, which is applied in the frequency domain, is used to parametrize impedance-based electrical models. Although the model following this characterization offers good results for different load profiles [6], the expensive equipment required, specifically the potentiostat or galvanostat, makes it less accessible.

However, due to our interest regarding the electrical behavior of the cell and considering the laboratory infrastructure, the characterization was achieved by using an electronic programable source–load configuration, where a modified version of the standard HPPC was deployed. Prior to this procedure, the capacity test and the open circuit voltage (OCV) test under different conditions were applied for the same cell [7].

Subsequently, the cell was modeled and simulated based on mathematical equations that describe the second order equivalent electric circuit (EEC), and, in the end, the real results and the simulated ones were compared for validation. For the comparison, two current profiles were used, one resulting from HPPC and one resulting following a simulation where an EV used the FTP-72 speed profile as a speed reference. Utilization of the second current profile [8–11] leads to more reliable results, because the current profile emulates a real current from the targeted applications. Thus, we can emphasize the novelty of the paper by stating that, here, the results are compared not only between the real and simulated ones, meaning the voltage response of the cell resulted from the HPPC, but also with the ones resulted following the FTP-72 driving cycle. This leads to a more realistic and dynamic testing where high intensity currents between 20 A and −100 A were throughput by the cell, therefore, verifying to the edge the reliability of the HPPC method.

## 2. The HPPC Method and the Developed EEC

Basically, in the HPPC method (as illustrated in Figure 1), positive and negative current pulses are applied to the cell at different amplitudes and SoC, and its voltage response is observed and recorded. The intensity of the amplitude is expressed as C-rate, which is a relative unit to the cell maximum capacity (e.g., for a 60 Ah cell, 1C is 60 A). The parameters of EEC are then computed based on the recorded voltage and considering the instant of its occurrence. This modified HPPC version started from a fully discharged cell, which then was charged with steps of 5%, where at every step, hybrid current pulses were applied. The duration of each pulse was 18 s, and the resting time between pulses was 15 min. In the original version of the HPPC presented in [12], steps of 10% of SoC are proposed, from 100% to 10% SoC, with 1 h resting time. At every step, a discharging pulse rated at 1C is applied for 10 s, followed by 40 s of resting time and then a charging pulse rated at 0.75C for 10 s. In [13], the author approached the same principle but for cells with significant lower capacity, and therefore, lower current intensity.

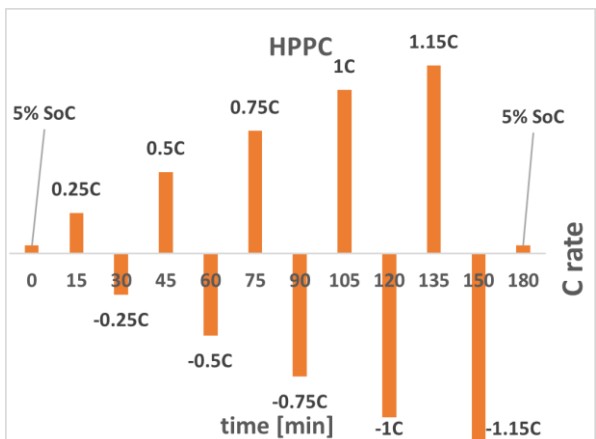

**Figure 1.** The HPPC testing sequence.

In [14], the authors elaborated a comprehensive study related to the comparison between one and two parallel circuits where it was concluded that for automotive application the last option is preferred. While applying the pulses, from the voltage response of the cell, an instant drop off voltage followed by an exponential curve is noticed. The EEC that reproduces the same voltage response can be assembled from a resistor (R0), called the ohmic resistor of the cell, in series with one or more branches of RC parallel circuits.

Here, the focus was on the circuit that contains two parallel RC networks, due to better performances than with a single RC network [12,15–17].

In Figure 2, there is an example of the voltage response of the cell for both types of pulses, charging and discharging, and the correspondent EEC. The first parallel RC circuit describes the activation polarization of the cell and it deals with the fast voltage response, while the last one is called the concentration polarization and is responsible for the slow voltage response.

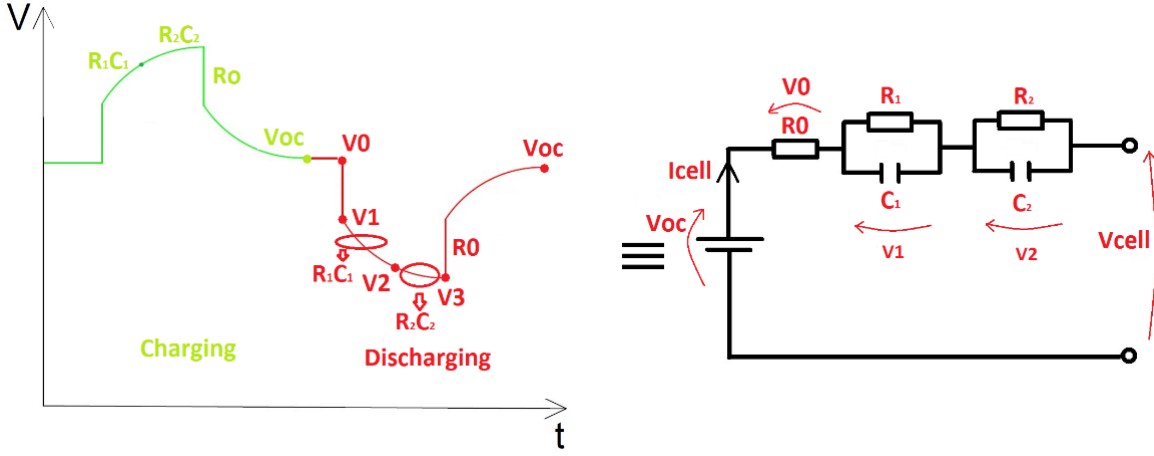

**Figure 2.** The voltage response of the cell and its second order EEC.

Using the first Kirchhoff law and a minor mathematical artifice, the drop off voltage for the parallel branches can be described by the differential Equations (1) and (2) [18].

$$\frac{dV_1}{dt} = \frac{I_{cell}}{C_1} - \frac{V_1}{R_1 C_1} \tag{1}$$

$$\frac{dV_2}{dt} = \frac{I_{cell}}{C_2} - \frac{V_2}{R_2 C_2} \tag{2}$$

Thus, the voltage at the cells terminals is described by the second Kirchhoff law in Equation (3).

$$V_{cell} = V_{oc} + (R_0 \times I_{cell} + V_1 + V_2) \tag{3}$$

where $V_{oc}$ is the open circuit voltage mentioned before.

The EEC parameter revelation begins once with the characterization process by recording the voltage at different instances. Thus, the first voltage measurement ($V_a$) is recorded prior to the current pulse. The second measurement ($V_b$) is triggered just after the beginning of the current pulse, 0.1 s later, in this case. The next ($V_c$) and the last one ($V_d$) are considered after 10 s and 17.9 s, respectively. According to Ohm's law, now the three circuit resistors values can be computed with Equations (4)–(6) and, afterwards, the capacitor values with (7) and (8) [19].

$$R_0 = \frac{\Delta V_1}{\Delta I} = \frac{V_a - V_b}{I_{cell}} \tag{4}$$

$$R_1 = \frac{\Delta V_2}{\Delta I} = \frac{V_b - V_c}{I_{cell}} \tag{5}$$

$$R_2 = \frac{\Delta V_3}{\Delta I} = \frac{V_c - V_d}{I_{cell}} \tag{6}$$

$$\tau_1 = R_1 C_1 \tag{7}$$

$$\tau_2 = R_2 C_2 \tag{8}$$

## 3. Experimental Setup

Prior to the cell characterization process, a suitable hardware configuration was arranged capable of executing the HPPC. The entire laboratory setup is shown in Figure 3. Here, the PC records the sent data by the real-time (RTs) control unit (dSpace MicroLab-Box), which is in charge with the control of the entire process. The positive current pulses are applied by the EA PS 8200-70 power supply, while the negative ones by an EA ELR 9500-90 electronic load. The communication between the control unit and the source/load is facilitated through CAN interface and voltage analogue signals. In addition, the cell voltage is measured directly by the RT unit from its leads, while the current is measured by high precision current transducers (LEM Ultrastab IT-200S).

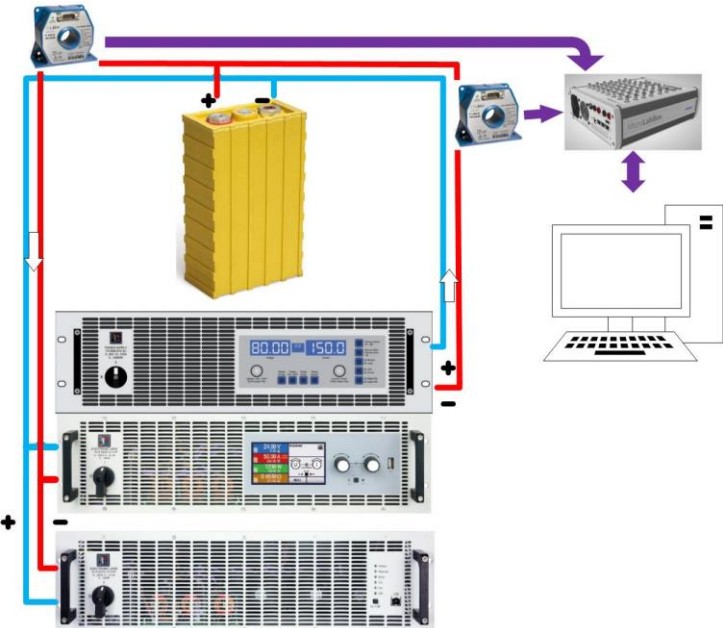

**Figure 3.** The hardware setup for HPPC.

The tested LFP battery cell parameters are presented in Table 1—where the most important ones are the voltage limits and the charging currents.

**Table 1.** Lithium cell parameters Winston FLP060AHA.

| Parameter | Unit | Value |
| --- | --- | --- |
| Nominal Voltage | V | 3.2 |
| Maximal charge voltage | V | 4 |
| Deep discharge voltage | V | 2.5 |
| Operating Voltage | V | 2.8–4 |
| Capacity | Ah | 67 |
| Max discharging current | A | 600 |
| Optimal discharging current | A | 30 |
| Max charging current | A | 180 |
| Optimal charging current | A | 30 |

Although the manufacturer claims the cell capacity at 60 Ah, the capacity test results show an enhanced capacity up to 67 Ah at 20 °C and 1C, thus, this was considered the rated value during the tests.

## 4. Characterization Results

The characterization process took approximatively one day and a half due to resting time between pulses and large number measurements. While the characterization proce-

dure ran, for each pulse, four voltage measurements were recorded. Of course, due to paper space constraints, Figure 4a only illustrates one step of 5% for the charging stage and hybrid pulses applied at 95% of SoC.

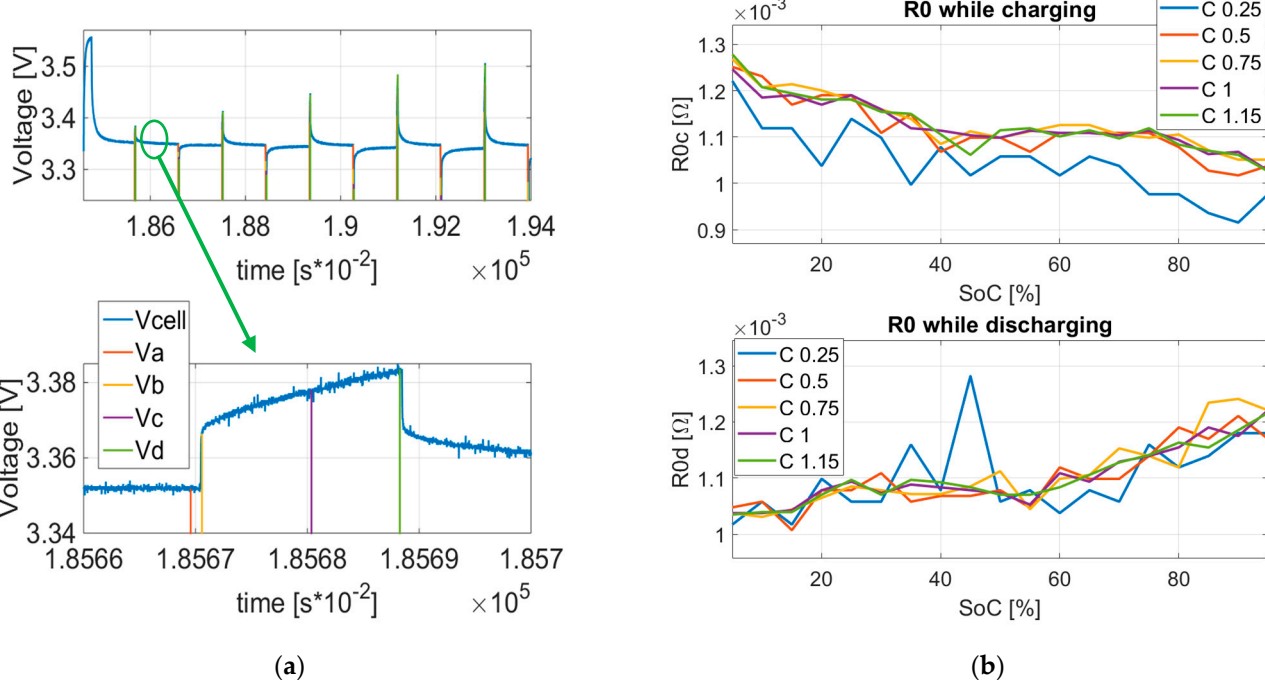

**Figure 4.** (**a**) The voltage response of the LFP cell during HPPC; (**b**) The ohmic resistance of the cell.

As previously mentioned, applying (4)–(8), we obtain the EEC parameters, with a resolution of 5%, between 5% and 95%. As shown in Figures 4 and 5, one can observe the parameter evolution for all five circuit elements under the charge and discharge operating regime.

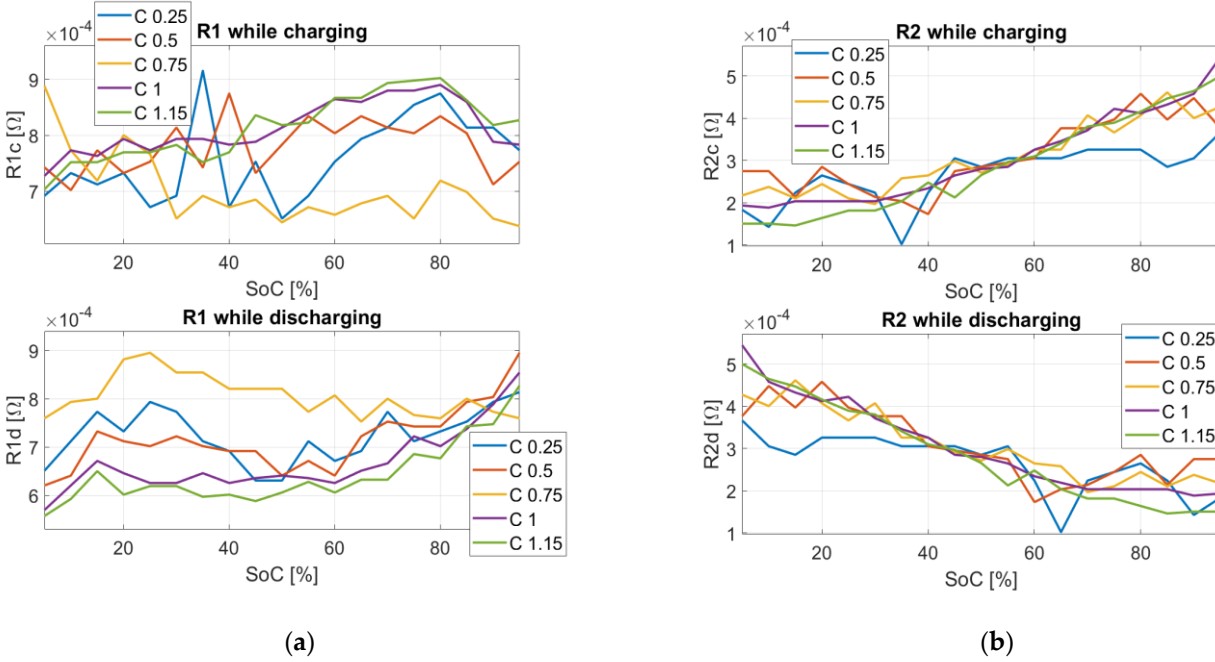

**Figure 5.** *Cont.*

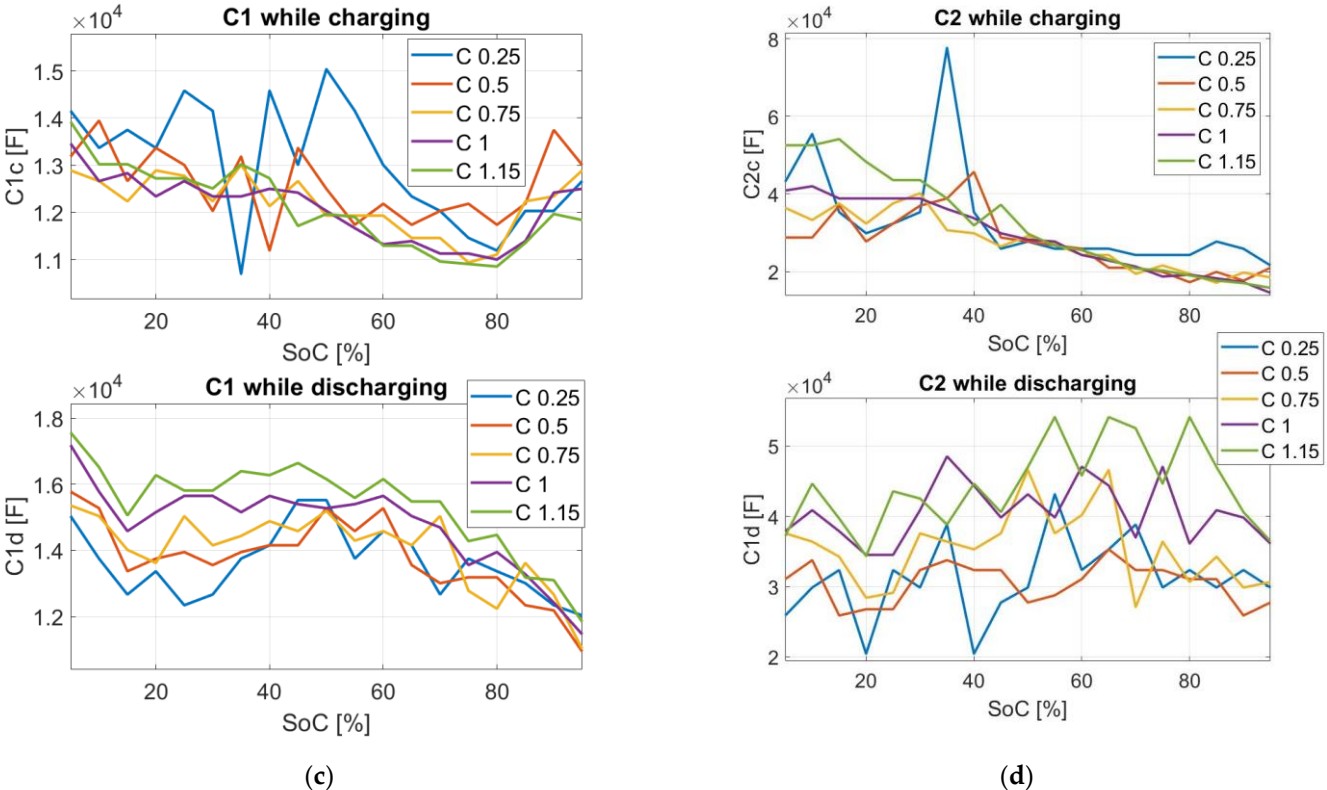

**(c)**　　　　　　　　　　　　　　　　　　　　　　　**(d)**

**Figure 5.** Evolution of the EEC parameters for (**a**) activation polarization resistor; (**b**) concentration polarization resistor, (**c**) activation polarization capacitor; (**d**) concentration polarization capacitor.

## 5. The Model Implementation

The model of the cell was implemented in MATLAB/Simulink software using Equations (1)–(3) for voltages and (13) for determining the SoC. Moreover, the resulted parameters and measured OCV were embedded using look-up tables, 2D for resistance and capacitance due to variation in the SoC and C-rate, while the OCV changed with SoC, thus, 1D was used. Additionally, for the OCV, when operating with a dynamic current profile, an artifice is required to be implemented due to the hysteresis characteristic of the voltage in the charging and discharging regime. Therefore, the OCV is modeled as a sum between the average OCV and hysteresis voltage, as seen in (9) [20]:

$$V_{oc} = V_{oc,avg} + V_h \tag{9}$$

where $V_{oc,avg}$ is computed as:

$$V_{oc,avg}(SoC) = \frac{1}{2}[V_{oc,ch}(SoC) + V_{oc,dis}(SoC)] \tag{10}$$

Hysteresis voltage $V_h$ is modeled by a differential equation, written as:

$$\frac{dV_h}{dt} = \beta I_{cell}[V_{oc,ch}(SoC) + V_{oc,dis}(SoC)] \tag{11}$$

Returning to the coefficient β, this has the expression:

$$\beta = \frac{1}{0.1 \times 3600 \times Q_{cell}} \tag{12}$$

where $Q_{cell}$ is the cell capacity.

In the end, the cell SoC results from:

$$SoC = \left(SoC_0 - \int \frac{I_{cell}}{3600 \times Q_{cell}}\right) \times 100 \tag{13}$$

## 6. The Obtained Results

The model reliability following the HPPC method is tested by comparing the real measured values with the simulated values for two dynamic current profiles, the HPPC and FTP-72 speed profile.

### 6.1. HPPC

For the first cycle, the current of the real cell was recorded and served as reference for the simulated model. The voltage response of the real cell and simulated one were compared and the relative error was computed. As illustrated in Figure 6, where the levels of the relative error are depicted in the second graph, the maximum error was 2.42%. As expected, this was found to be close to 90% of the SoC. Generally, the maximum error is obtained at the SoC extremities. Moreover, in order to check the overall model performance, expressed through root mean square error (RMSE), this parameter was also computed and situated at 0.48%.

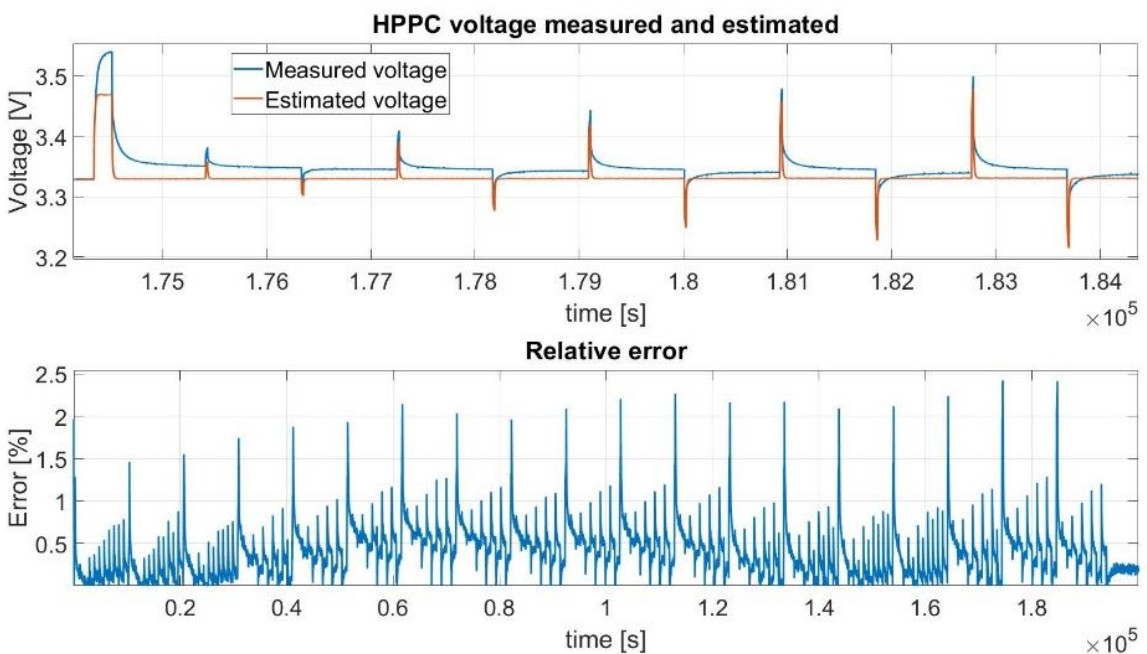

**Figure 6.** The tested and simulated HPPC profile.

### 6.2. FTP-72

The last method has a current profile resulting from a launched simulation in Amesim software where an EV follows the FTP-72 speed profile. The current profile of the battery was recorded, scaled to our cell performances, and then applied to the cell at different SoC as 100%, 70%, and 30%. The idea is to have a real current profile, not just pulses, reflecting the real exploitation regime. After each FTP-72 cycle, the cell was completely discharged, fully charged, and then discharged again at 1C-rate until it reached the desired SoC. This process was realized in order to eliminate the impact of the historic current profile over the OCV, thus, recreating the same conditions when the OCV-SoC characterization step was performed, leading to more relevant and comparable results [21–23].

In the left side of, Figure 7a, the voltage response is depicted for both cells, tested and simulated, where, at 100% SoC, a maximum relative error of 4.78% is noticed.

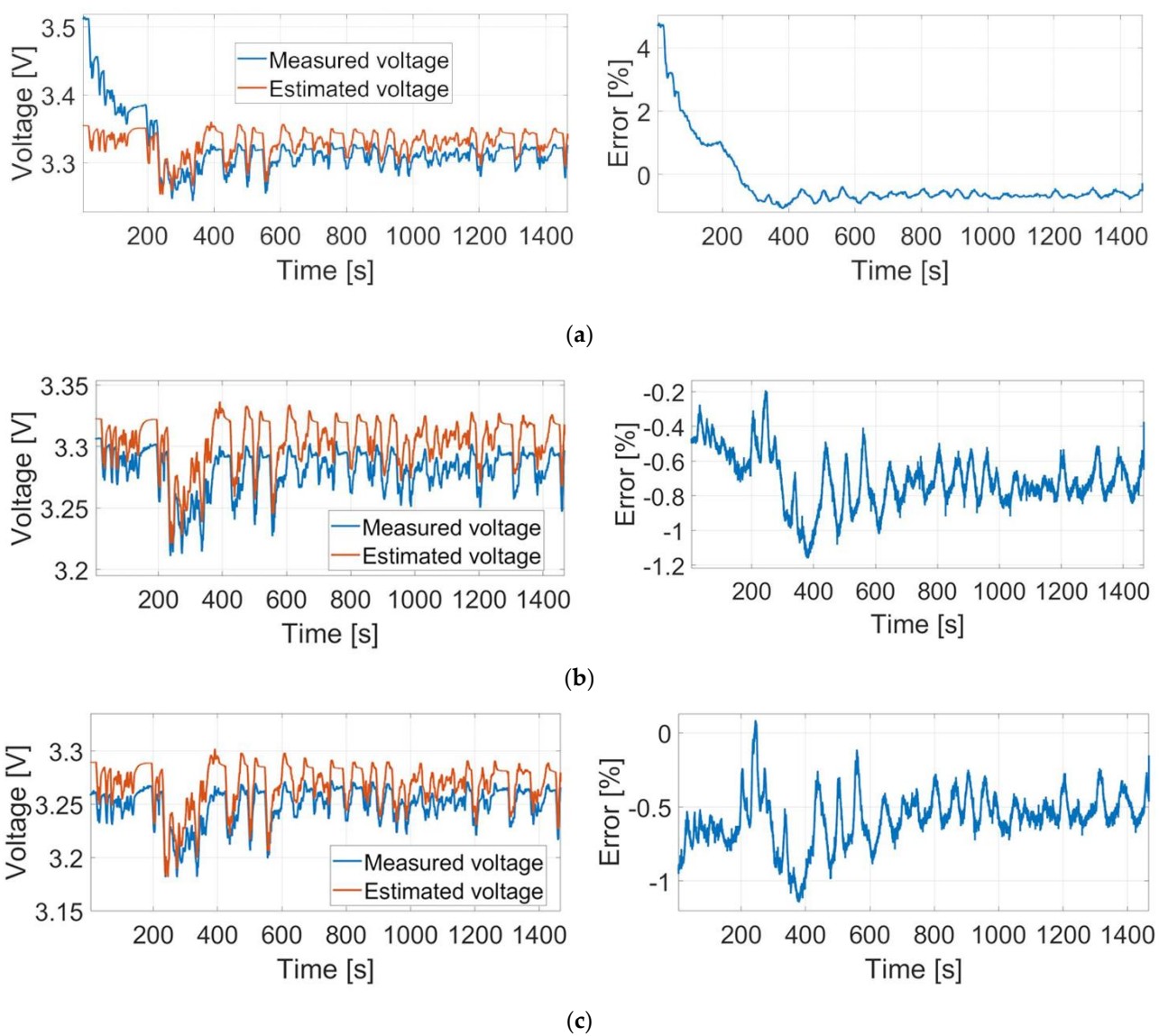

**Figure 7.** The real and simulated HPPC profile applied at (**a**) 100% of the SoC; (**b**) 70% of the SoC; (**c**) 30% of the SoC.

When the same current profile is put through the cell at 70% SoC, as seen in Figure 7b, the results are improved, offering a maximum error of −1.15%. Thus, in this case, the model overestimates the cell voltage. In the end, the test was made at 30% SoC; the results are presented in Figure 7c. Again, as previously mentioned, the model underestimates the voltage, as the error maintains in negative plan with a small exception. Here, the maximum error is −1.14%. Tests under 30% SoC were not made because of the risk for exceeding the minimum threshold voltage of the cell, which would compromise its chemistry.

## 7. Conclusions

In this work, the results of the HPPC for a LFP cell with a considerable capacity of 67 Ah were presented, when tests were employed under ambient temperature and relative high level of currents up to 70 A. The entire process was employed by a hardware configuration containing programmable source–load equipment controlled by an FPGA based real-time platform. Following the method, the results were implemented into a Simulink model and described with equations of the second order obtained from an EEC. The results from the real cell and the simulated cell were compared in order to verify the

model's reliability, using two dynamic current profiles. The first resulted from the HPPC method, while the other one was based on the FTP-72 speed profile. The compared results for the first case were acceptable, offering a maximum relative error for the HPPC cycle of 2.42%, while the overall error settled at 0.48% RMSE. In the end, the last three cases where the speed profile was applied, the maximum error level for 100% SoC was 4.78%, $-1.15\%$ for 70 SoC, and 1.14% for 30% SoC.

It can be concluded that this characterization method offers reliable parameters for subsequent simulation of the cell based on second order EEC where the accepted error thresholds are on par with previously mentioned ones.

**Author Contributions:** Conceptualization, R.C.N. and D.F.; methodology, R.C.N.; validation, R.C.N. and D.F.; formal analysis, R.C.N. and D.F.; investigation, R.C.N.; resources, D.F.; writing—original draft preparation, R.C.N.; writing—review and editing, R.C.N. and D.F.; funding acquisition, D.F. All authors have read and agreed to the published version of the manuscript.

**Funding:** This work was partially supported by a grant from the Romanian Ministry of Research and Innovation, CCCDI-UEFISCDI, project number PN-III-P1-1.2-PCCDI-2017-0776/No. 36 PC-CDI/15.03.2018, within PNCDI III. This work was partially supported by the Project "Entrepreneurial competences and excellence research in doctoral and postdoctoral programs-ANTREDOC", No. 56437/24.07.2019, project co-funded by the European Social Fund.

**Data Availability Statement:** Not applicable.

**Conflicts of Interest:** The authors declare no conflict of interest.

## Nomenclature

| | |
|---|---|
| LFP | lithium iron phosphate |
| HPPC | hybrid pulse power characterization |
| EV | electric vehicle |
| EVSE | electric vehicle supply equipment |
| FTP-72 | federal test procedure driving cycle |
| EU | European union |
| EIS | electrochemical impedance spectroscopy |
| RC | resistor–capacitor |
| OCV | open circuit voltage |
| A | amperes |
| C | is a measure of the rate at which a battery is discharged relative to its capacity |
| EEC | electrical equivalent circuit |
| RT | real-time |
| CAN | controller area network |
| 1D, 2D | one dimension, two dimensions |
| RMSE | root mean square error |
| $C_{1,2}$ | the capacitances for the parallel branches, 1 and 2, of the battery's equivalent circuit |
| $I_{cell}$ | the current passing one cell |
| $Q_{cell}$ | the Ah capacity for one cell |
| $R_0$ | ohmic resistor for one cell |
| $R_{1,2}$ | the resistances for the parallel branches, 1 and 2, of the battery's equivalent circuit |
| SoC | the state-of-charge |
| $V_0$ | the drop off voltage for the ohmic resistor |
| $V_{1,2}$ | the drop off voltage for the parallel branches, 1 and 2, of the battery's equivalent circuit |
| $V_{a,b,c,d}$ | voltage measured at different moments of time during each pulse of HPPC |
| $V_{oc,avg}$ | the open-circuit average voltage |
| $V_{cell}$ | the voltage for one cell |
| $V_h$ | the hysteresis voltage |
| $V_{oc}$ | the open-circuit voltage |
| $V_{oc,ch}$ | the charging open-circuit voltage |
| $V_{oc,dis}$ | the discharging open-circuit voltage |
| $\beta$ | hysteresis coefficient |
| $\tau_{1,2}$ | time constants for the parallel branches, 1 and 2, of the battery's equivalent circuit |

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
