# Peer review of "Lithium-Ion Cell Characterization, Using Hybrid Current Pulses, for Subsequent Battery Simulation in Mobility Applications"

_processes, doi:10.3390/pr10102108_

Round 1

Reviewer 1 Report

In this manuscript, the authors propose and evaluate a characterization method for a Lithium Iron Phosphate (LFP) pouch cell, which applies hybrid current pulses called Hybrid Power Pulse Characterization (HPPC). And the reliability of the mathematical model following the HPPC method is verified by tests. However, some major issues should be properly addressed before publication. The specific comments/questions are listed as below:

1. The HPPC method used for characterization and simulation of Li-ion batteries has been reported previously, such as DOI:10.1109/ECCE-Asia49820.2021.9479017. The novelty or advancement of this manuscript should be further emphasized compared with previously reported studies.

2. In the Introduction section, the discussion on air quality and electric vehicle sales in the first two paragraphs is unnecessary. It is suggested to simplify the description. The authors should introduce more about the modeling technology of battery and its parameter estimation methods.

3. In part 2, the authors stated that “Here, the focus was on the circuit which contains two parallel RC networks, due to better performances than with a single RC network.” Please provide references to support this view and illustrate the advantages of the two parallel RC networks.

4. The test results of the model in this manuscript are lack of result comparison. The authors should compare the simulation results of the model following the HPPC method with those of other relevant simulation models to highlight the accuracy and reliability of this method.

5. There is no simulation result of this model on the state of charge (SoC) of the batteries in this paper. Please illustrate whether this model can accurately simulate the SoC of the batteries.

6. In order to improve the quality of the manuscript, some types errors should be corrected. For example, the operating voltage values “2.8÷4” in Table 1 should be corrected to “2.8-4”. On page 7, the authors stated that “using equations (1)-(3) for voltages and (11) for determining the SoC”. In this sentence, “(11)” should be changed to “(13)”. In addition, the serial numbers of each section of the article should be sorted correctly.

Author Response

pouch cell, which applies hybrid current pulses called Hybrid Power Pulse Characterization (HPPC). And the reliability of the mathematical model following the HPPC method is verified by tests. However, some major issues should be properly addressed before publication. The specific comments/questions are listed as below:

Answer : thank you very much for your effort in revising our manuscript. We have careffuly investigated your raised issues and critics and we have addresses all of them. Here bellow, our answers, in blue, for each raised critic.

  1. The HPPC method used for characterization and simulation of Li-ion batteries has been reported previously, such as DOI:10.1109/ECCE-Asia49820.2021.9479017. The novelty or advancement of this manuscript should be further emphasized compared with previously reported studies.

Answer 1:

As you mentioned, the HPPC method was presented before and in the suggested paper we can find a good example in what concerns method description and presented results. In order to emphasize the originality of the paper, the following paragraph was added at the end of the introduction section, at page 2:

Thus, we can emphasize the novelty of the paper by stating that here the results are compared not only between the real and simulated ones, meaning the voltage response of the cell resulted from the HPPC, but also with the ones resulted following the FTP-72 driving cycle. This leads to a more realistic and dynamic testing where high intensity currents between 20 A and -100 A were throughput by the cell, therefore verifying to the edge the reliability of the HPPC method.

  1. In the Introduction section, the discussion on air quality and electric vehicle sales in the first two paragraphs is unnecessary. It is suggested to simplify the description. The authors should introduce more about the modeling technology of battery and its parameter estimation methods.

Answer 2:

The paragraphs were intended to emphasize the motivation behind this research project and of this paper, but, as suggested, this might be irrelevant, therefore we removed them in the newer version.

Regarding the modeling technology of the battery and its parameter estimation methods, some information is already given with this respect in paragraph 2 from section 1. Nevertheless, we have added the following paragraph, at page 2:

“For example, in [4] the authors evaluated the existing characterization methods for auto-motive applications with respect to the main international battery test standards, combining them into one procedure. Here, the HPPC current profile was used and the parameters were estimated for a RC parallel network followed by a RC series network, but a comparison between the real and a simulated cell model is missing. Furthermore, in [5], Electrochemical Impedance Spectroscopy (EIS) technique, which is applied in the frequency domain, it is used to parametrize impedance-based electrical models. Although the model following this characterization offers good results for different load profiles, [6], the expensive equipment required as Potentiostat or Galvanostat defines it as not too accessible.”’

  1. In part 2, the authors stated that “Here, the focus was on the circuit which contains two parallel RC networks, due to better performances than with a single RC network.” Please provide references to support this view and illustrate the advantages of the two parallel RC networks.

Answer 3:

We have added the reference which supports the statement – please check the info at page 2:

In [14] the authors had elaborated a comprehensive study related on the comparison between one and two parallel circuits, where it is concluded that for automotive application the last option is preferred.

  1. The test results of the model in this manuscript are lack of result comparison. The authors should compare the simulation results of the model following the HPPC method with those of other relevant simulation models to highlight the accuracy and reliability of this method.

Answer 4:

A comparison between the performances of different EEC topologies and second order EEC or between HPPC and another characterization method (e.g. EIS) presents much interest for us and we already a made comparison between the simulated and measured results, given in Fig.6 & 7.

  1. There is no simulation result of this model on the state of charge (SoC) of the batteries in this paper. Please illustrate whether this model can accurately simulate the SoC of the batteries.

Answer 5:

In the presented model the SoC was calculated using coulombic method, where the capacity of the cell was considered constant. Therefore, this model can only accurately simulate the SoC of the cell considering one operating point. In reference [8] we presented a study where we approached the cell capacity for multiple operating points depending on the current, but the difference was below 1 Ah. In the future, we plan to enhance the model performances by considering the capacity, temperature and the cell state of health.

  1. In order to improve the quality of the manuscript, some types errors should be corrected. For example, the operating voltage values “2.8÷4” in Table 1 should be corrected to “2.8-4”. On page 7, the authors stated that “using equations (1)-(3) for voltages and (11) for determining the SoC”. In this sentence, “(11)” should be changed to “(13)”. In addition, the serial numbers of each section of the article should be sorted correctly.

Answer 6:

We have entirely revised the paper, both, in its content, as well as in its format, and we have improved the quality of figures, text, and we have corrected all the aforementioned issues.

Reviewer 2 Report

I suggest the publication of the paper after minor corrections.

Increase and unify the font size of figures 4 and 5. In some of these, improve the resolution.

Author Response

I suggest the publication of the paper after minor corrections.

Increase and unify the font size of figures 4 and 5. In some of these, improve the resolution.

Answer: thank you very much for your effort in reviewering our article proposal.

The manuscript was entirely revised in its content.

Also, we have re-ploted, for quality improvement, Figures 4 & 5, for a better lisibility, as well as other figures. Also, the manuscript was entirely revised in its format.

Round 2

Reviewer 1 Report

The manuscript is advised to be accepted since all the problems have been addressed.
